## Research Article

water insecurity; LGBTQ; India; Thailand; urban; mental health

**Corresponding author:**
Carmen H. Logie;
Email: carmen.logie@utoronto.ca

# Associations between water insecurity and mental health outcomes among lesbian, gay, bisexual, transgender and queer persons in Bangkok, Thailand and Mumbai, India: Cross-sectional survey findings

Carmen H. Logie[1,2,3,4] (iD), Peter A. Newman[1], Zerihun Admassu[1], Frannie MacKenzie[1], Venkatesan Chakrapani[5], Suchon Tepjan[6], Murali Shunmugam[5] and Pakorn Akkakanjanasupar[7]

[1]Factor Inwentash Faculty of Social Work, University of Toronto, Toronto, ON, Canada; [2]United Nations University Institute for Water, Environment, and Health, Hamilton, ON, Canada; [3]Women's College Research Institute, Women's College Hospital, Toronto, ON, Canada; [4]Centre for Gender & Sexual Health Equity, Vancouver, BC, Canada; [5]Centre for Sexuality and Health Research and Policy (C-SHaRP), Chennai, India; [6]VOICES-Thailand Foundation, Chiang Mai, Thailand and [7]Department of Educational Policy, Management, and Leadership, Faculty of Education, Chulalongkorn University, Bangkok, Thailand

## Abstract

**Background:** Water insecurity disproportionally affects socially marginalized populations and may harm mental health. Lesbian, gay, bisexual, transgender and queer (LGBTQ) persons are at the nexus of social marginalization and mental health disparities; however, they are understudied in water insecurity research. Yet LGBTQ persons likely have distinct water needs. We explored associations between water insecurity and mental health outcomes among LGBTQ adults in Mumbai, India and Bangkok, Thailand.

**Methods:** This cross-sectional survey with a sample of LGBTQ adults in Mumbai and Bangkok assessed associations between water insecurity and mental health outcomes, including anxiety symptoms, depression symptoms, loneliness, alcohol misuse, COVID-19 stress and resilience. We conducted multivariable logistic and linear regression analyses to examine associations between water insecurity and mental health outcomes.

**Results:** Water insecurity prevalence was 28.9% in Mumbai and 18.6% in Bangkok samples. In adjusted analyses, in both sites, water insecurity was associated with higher likelihood of depression symptoms, anxiety symptoms, COVID-19 stress, alcohol misuse and loneliness. In Mumbai, water insecurity was also associated with reduced resilience.

**Conclusion:** Water insecurity was common among LGBTQ participants in Bangkok and Mumbai and associated with poorer well-being. Findings signal the importance of assessing water security as a stressor harmful to LGBTQ mental health.

## Impact statement

Water insecurity – disproportionately impacting socially marginalized communities – affects two billion people and is an urgent global health concern. Water insecurity is linked with poorer mental health outcomes through complex pathways, including distress, uncertainty, reduced self-efficacy and relationship conflict. Yet water insecurity and linkages to mental health are understudied with lesbian, gay, bisexual, transgender and queer (LGBTQ) persons. Structural violence, the ways in which social structures (e.g., political, economic religious) limit persons from realizing their potential, contributes to social marginalization of LGBTQ persons and poorer mental health outcomes. India and Thailand are LMICs affected by water insecurity, where there are also notable LGBTQ mental health disparities. In these two contexts, LGBTQ persons may also, as a result of social, educational and economic marginalization, live in housing with insufficient water, sanitation and hygiene (WASH) access. We conducted the first study to examine water insecurity and associations with mental health among LGBTQ persons in Mumbai, India and Bangkok, Thailand. We found that water insecurity is widespread – affecting approximately one-fifth of LGBTQ participants in Bangkok (18.6%) and one-quarter (28.9%) in Mumbai – and was associated with anxiety symptoms, depression symptoms, loneliness, alcohol misuse, COVID-19 stress and resilience. This points to water insecurity as a possible additional minority stressor that can harm LGBTQ mental health in LMICs that warrants further investigation and intervention. Our work signals the need to address knowledge gaps regarding

the experiences of water insecurity among LGBTQ persons in Bangkok and Thailand. LGBTQ persons may have distinct water needs, as they may face unique challenges in accessing WASH due to larger contexts of stigma. With ongoing LGBTQ urbanization, increases in climate-related water insecurity and preexisting LGBTQ social marginalization and mental health challenges, researchers can include sexual orientation and gender identity in WASH research and water insecurity in mental health research with LGBTQ persons.

## Introduction

Water insecurity – constrained access to reliable, safe and sufficient water – is a widespread issue affecting two billion people who lived with no safely managed water services in 2020 (JMP, 2021). Water insecurity is associated with poorer mental health across diverse global regions (Cooper-Vince et al., 2018; Maxfield, 2020; Mushavi et al., 2020; Stoler et al., 2020; Wutich et al., 2020). The pathways accounting for this relationship include distress, worry, shame, uncertainty, reduced self-efficacy and independence and enhanced relationship conflict (Cooper-Vince et al., 2018; Mushavi et al., 2020; Wutich et al., 2020). Globally, water insecurity disproportionately affects socially marginalized populations (Wutich et al., 2022), including persons with lower socioeconomic status (Meehan et al., 2020; Stoler et al., 2020). Yet lesbian, gay, bisexual, transgender and queer persons (LGBTQ) experience social marginalization across global regions (Poteat et al., 2021) – contributing to mental health challenges that may be exacerbated in the COVID-19 pandemic (Ormiston and Williams, 2022) – and are understudied in water insecurity research (Benjamin and Hueso, 2017; Wutich, 2020). This is a key gap, as social marginalization contributes to a disproportionately high rate of mental health concerns among LGBTQ persons globally (Ayhan et al., 2020; Mongelli et al., 2019; Scheim et al., 2020; Williams et al., 2021). Mental health challenges among LGBTQ persons in India and Thailand were exacerbated in the COVID-19 pandemic due to a constellation of factors, including: pandemic-related closures of LGBTQ support services; loss of employment, including among sex workers; increased poverty and subsequent food and housing insecurity; and mobility restrictions that reduced access to social support systems (Chakrapani et al., 2022, 2022; Newman et al., 2021; Yasami et al., 2023). Thus, it is urgent to understand this nexus of water insecurity and mental health among LGBTQ persons in India and Thailand to advance health and well-being.

Linkages between water insecurity and depression are documented among non-LGBTQ populations in diverse LMIC contexts such as Uganda (Cooper-Vince et al., 2018; Logie et al., 2023; Mushavi et al., 2020), Kenya (Miller et al., 2021), Haiti (Brewis et al., 2019), Vietnam (Vuong et al., 2022) and Nepal (Aihara et al., 2016). There is increasing attention to water insecurity-related mental health stressors in urban LMIC settings (Abadi et al., 2020). For instance, studies with cisgender women in Odisha, India (Hulland et al., 2015) reported how insufficient water access exacerbated psychosocial stress regarding meeting sanitation needs. LGBTQ persons are disproportionately affected by poor mental health outcomes in India and Thailand compared with general populations – with an estimated five-fold higher odds of depression and alcohol dependence in India (Chakrapani et al., 2017; Wandrekar and Nigudkar, 2020; Wilkerson et al., 2018) and 10-times higher depression in Thailand (Kittiteerasack et al., 2021; Kongsuk et al., 2013). LGBTQ persons experience pervasive stigma and discrimination in these contexts, as in other global regions, that contributes to these mental health disparities (Chakrapani et al., 2019; Ojanen et al., 2019) through distal (e.g., discrimination) and proximal (e.g. internalized stigma) stressors, as conceptualized in

the minority stress model (Kittiteerasack et al., 2021; Logie et al., 2012; Meyer, 1995). India and Thailand both lack the following: broad protection against discrimination based on sexual orientation; criminal liability for offences committed on the basis of sexual orientation; prohibition of incitement to hatred, violence or discrimination based on sexual orientation; same-sex marriage recognition and partnership recognition for same-sex couples ("Home | ILGA World Database", 2023; Mendos and ILGA World, 2019) – limiting access to legal or other recourse in the face of stigma and discrimination and constraining access to relational benefits with partners (Newman et al., 2021; Reid et al., 2022). This lack of legal protection converges with the limited access to broad mental health services and trained mental health providers competent with LGBTQ issues in these contexts (Chakrapani et al., 2020; Ojanen et al., 2019).

India and Thailand are LMICs also affected by water insecurity (JMP, 2021). Urbanization combined with limited and aging water infrastructure results in urban water shortages in both Bangkok (Chapagain et al., 2022; Edelman, 2022) and Mumbai (Subbaraman and Murthy, 2015), including insufficient and unsafe water supplies. Basic hygiene facilities – access to soap and water for handwashing at home – was reported by 68% of persons in India and 85% of persons in Thailand (JMP, 2021). Globally there is also urbanization of LGBTQ persons (Ayoub and Kollman, 2021; Oswin, 2015), in part due to the formation of LGBTQ subcultures and social movements in urban spaces and stigma in places of origin (Liu et al., 2018). Despite the convergence of contexts of water insecurity, urbanization and LGBTQ mental health disparities in India and Thailand, there remains a scarcity of data on water insecurity and related mental health outcomes among LGBTQ persons in these contexts.

Importantly, this knowledge gap regarding LGBTQ persons' experiences of water insecurity extends across global contexts (Benjamin and Hueso, 2017; Brewis et al., 2023; Wutich, 2020). The discussion of gender in the water, sanitation and hygiene (WASH) sector focuses on cisgender (non-transgender) persons and the gender binary (women, men) (Benjamin and Hueso, 2017; Brewis et al., 2023; Wutich, 2020). Yet LGBTQ persons' gender roles, norms, practices and family configurations expand beyond and often conflict with cisgender, heterosexual gender dynamics (Tannenbaum et al., 2016). Thus, a better understanding water insecurity experiences among LGBTQ populations is urgently needed (Brewis et al., 2023). For instance, a Nigerian study with LGBTQ persons found water insecurity was associated with living with a man, transactional sex and food insecurity (Hamill et al., 2023), and Brewis et al. describe how this study "highlights the intersectional nature of gender and sexual identities in creating risks for water insecurity" (p. 6) (Brewis et al., 2023). The limited research that has considered LGBTQ persons in relation to WASH is focused on sanitation, in particular, access to toilets among transgender persons (Benjamin and Hueso, 2017; Boyce et al., 2018; Moreira et al., 2021). For instance, in India and Nepal, literature documents transgender persons' experiences of harassment, abuse and violence when trying to use men's or women's toilets as well as denial of toilet access (Benjamin and Hueso, 2017;

Boyce et al., 2018). There remains a dearth of information on water access and water security among LGBTQ persons more broadly.

Structural violence is a conceptual framework that describes how the social world and its structures (e.g., political, economic legal) are organized in ways that cause harm, injury and ultimately prevent persons from realizing their potential (Farmer et al., 2006; Galtung, 1969). This framework was applied to understand how direct and indirect violence experienced by sexually diverse men in India, in legal, community, family and healthcare contexts, shape HIV vulnerability (Chakrapani et al., 2007). Structural violence may be relevant to contextualizing water insecurity risk and harms with LGBTQ persons. For instance, LGBTQ stigma and discrimination limits access to education, employment and housing opportunities, resulting in an overrepresentation of LGBTQ persons living in poverty (Badgett, 2012; Lee et al., 2019; SOGI Task Force and Dominik Koehler, 2015), often in resource-constrained environments such as in slums and informal settlements with WASH insecurity in India (Barik and Pattayat, 2022; Jadav and Chakrapani, 2023) and Thailand (Ojanen et al., 2019, 2019). Better understanding how water insecurity, an indicator of social marginalization and thus a form of structural violence, affects mental health among LGBTQ persons can inform health research, policy and practice at structural levels to advance LGBTQ persons' WASH needs as a human right (Heller, 2019) and individual levels to address immediate mental health needs.

Associations between water insecurity and mental health outcomes remain understudied with LGBTQ persons in LMICs, despite this population's noted experiences of social marginalization and health inequities. This paper aims to address this knowledge gap by examining the associations between water insecurity and mental health outcomes among LGBTQ persons in urban contexts in India (Mumbai) and Thailand (Bangkok).

## Methods

### Design

We present baseline survey data analyses for a two-site, parallel waitlist-controlled randomized controlled trial, which aimed to test the efficacy of an eHealth intervention to increase COVID-19 knowledge and protective practices and decrease pandemic-related psychological distress among LGBTQ adults (#SafeHandsSafeHearts) (Newman et al., 2021). eHealth refers to the ways in which digital technologies (e.g., mobile phones) are used to: monitor and track health and provide health information; communicate with health professionals and collect and manage health data (Shaw et al., 2017).

### Recruitment

Participants were recruited online using convenience sampling with e-flyers and messages via listservs and social media accounts of community-based organizations and clinics providing services to sexual and gender minority populations, LGBTQ e-groups and a study website. Potential participants were directed, if interested, to click a link to an online screening and baseline questionnaire.

### Procedures

Eligibility criteria were (1) age ≥18 years; (2) self-identify as cisgender lesbian or bisexual woman or woman who has sex with women; cisgender gay or bisexual man or man who has sex with men; or transfeminine, transmasculine or gender nonbinary individual; (3) resident in Mumbai/Thane area or Bangkok Metropolitan Area; (4) able to understand Hindi or Marathi (Mumbai) or Thai language (Bangkok) and (5) able to understand and willing to provide informed consent.

Screening for eligibility was conducted online, after which potential participants were presented with an online informed consent form. Individuals were instructed to contact the study coordinator if they wished to ask any questions or request clarifications before providing consent, and were provided with an email address and phone number, as well as a unique participant identifier to sign-in to the questionnaire. After providing consent, participants were linked to the baseline questionnaire.

Ethical approval was obtained from the Research Ethics Boards of the University of Toronto (39769), the Humsafar Trust (HST-IRB-S1-06/2021) and Chulalongkorn University (272/64). The trial is registered at ClinicalTrials.gov NCT04870723.

### Measures

#### Exposure

To assess water insecurity, we used four items of the Household Water Insecurity Experiences (HWISE) scale (Young et al., 2019) (Cronbach's alpha=0.85 in this study). Responses to items include never (0 times; score=0), rarely (1–2 times; score=1), sometimes (3–10 times; score=2), often (11-20 times; score=3) and always (more than 20 times; score=3). The total score range is 0–12, where higher scores indicate higher levels of household water insecurity. A household experiencing half of the four HWISE items 'sometimes' in the past 4 weeks would be considered water insecure. Therefore, we used a HWISE scale score of 4 or higher as the cut-off for household water insecurity.

#### Mental health outcomes

We assessed anxiety, depression, loneliness, alcohol use, COVID-19 stress and resilience. To assess anxiety symptoms, we used the generalized anxiety disorder (GAD-2) screening tool that ranged from 0 to 6; we used the cut-off of 3 and above for GAD symptoms (Spitzer et al., 2006) (Cronbach's alpha= 0.78 in this study). The two items queried how often over the last 2 weeks a person was bothered by feeling nervous, anxious or on edge and not being able to stop or control worrying.

We used the two-item Patient Health Questionnaire screening for depression symptoms (Cronbach's alpha=0.75 in this study). The two items assess how much participants had experienced depressive mood and anhedonia in the past 2 weeks. Individuals' responses for the two items were summed, and a cut-off score of 3 was used as the cut-off point for screening depression symptoms (Kroenke et al., 2001).

The Three-Item Loneliness Scale, developed from the R-UCLA Loneliness Scale, was used to measure perceptions of loneliness (Hughes et al., 2004) (Cronbach's alpha=0.84 in this study). Respondents were asked how often they felt that they lacked companionship, were left out, and were isolated from others, on a 3-point Likert scale (1 'hardly ever' to 3 'often'). Individuals' responses were summed, with higher scores indicating greater loneliness (range: 3–9). In this study, a cut-off score of above 5 was used to define loneliness (Matthews et al., 2022).

We used AUDIT-C to screen for alcohol misuse (Bush, 1998) (Cronbach's alpha=0.73 in this study). The three items assess any alcohol use in the past year, the number of drinks on a typical day of

drinking in the past year, and the frequency of consuming 7 or more drinks on one occasion in the past year. AUDIT-C scores were summed for a possible score of 0–12. We coded men (transgender and cisgender inclusive) reporting scores of 4 or higher as positive for alcohol misuse and women (transgender and cisgender inclusive) reporting scores of 3 or higher as positive for alcohol misuse.

To assess COVID-19 stress we used a 9-item COVID-19 Stress Scale (Taylor et al., 2020) (Cronbach's alpha=0.90 in the current study). Items were scored on a 5-point Likert scale (1 'never' to 5 'always'). Items were summed to create a total score and the score ranged from 9 to 45, with a higher total score indicating higher stress levels.

Resilience was measured using the four-item Response to Stressful Experiences Scale (Johnson et al., 2011) (Cronbach's alpha=0.87 in this study). Item scores range from 1 (not at all like me) to 5 (exactly like me). Items were summed to create a total score with higher scores indicating more resilient responses to stressful events.

Covariates assessed included age (continuous), highest education level, sexual orientation, gender identity, losing one's job due to COVID-19-related factors, current employment status and self-reported HIV serostatus.

## Analysis

Descriptive statistics were performed to describe sociodemographic characteristics using the mean and standard deviation (SD) of continuous variables and frequencies and proportions of categorical variables. The prevalence of water insecurity and mental health outcomes among participants was calculated using cross-sectional data from Mumbai and Bangkok samples.

Binary logistic regression models for binary categorical outcomes, and linear regression models for continuous outcomes, were used to examine associations between water insecurity and mental health outcomes (depression symptoms, anxiety symptoms, COVID-19 stress, resilience, loneliness alcohol use). Multivariate logistic and linear regression models were conducted to examine independent associations between water insecurity and mental health outcomes after adjusting for all variables that were associated with mental health outcomes in bivariate analyses. Estimates of the strengths of associations were calculated using adjusted odds ratios (AOR) and beta coefficients (β) with 95% confidence intervals (95% CI). Results were considered significant when the probability value was p≤0.05. All statistical analyses were performed using STATA version 17.0 software.

## Results

### Sample characteristics

From August 2021 to February 2022, 650 participants completed the baseline survey. Table 1 summarizes the sociodemographic characteristics of the full sample population (n=650) as well as by location for Mumbai, India (n=290; 44.6%) and Bangkok, Thailand (n=360; 55.4%). Most of the population (64.2%) identified as cisgender, whereas the rest (35.8%) identified as transgender. The mean age of the sample was 30.7±7.5 years, most completed college/university level education (54.5%) and were employed (70.7%). Of the sample population, 7.4% self-reported as HIV positive and nearly one-quarter reported water insecurity (23.2%). About one-third of the population reported depressive symptoms (28.6%), anxiety symptoms (28.5%) and/or alcohol misuse (31.4%) and more than half experienced loneliness (53.1%) and/or COVID-

**Table 1.** Characteristics of study participants in Mumbai, India and Bangkok, Thailand (N = 650)

| Characteristics | Mumbai: n = 290 N (%) | Bangkok: n = 360 N (%) | Total: N = 650 N (%) |
|---|---|---|---|
| Age (Mean/SD) | 29.8 (7.1) | 31.5 (7.8) | 30.7 (7.5) |
| Gender | | | |
| Cisgender woman | 50 (17.2) | 121 (33.6) | 171 (26.3) |
| Cisgender man | 124 (42.8) | 122 (33.9) | 246 (37.8) |
| Transgender woman/ transfeminine | 80 (27.6) | 90 (25) | 170 (26.2) |
| Transgender man/transmasculine | 36 (12.4) | 27 (7.5) | 63 (9.7) |
| Gender identity | | | |
| Cisgender | 174 (60) | 243 (67.5) | 417 (64.2) |
| Transgender | 116 (40) | 117 (32.5) | 233 (35.8) |
| Highest level of completed education | | | |
| None/primary/elementary | 24 (8.2) | 15 (4.2) | 39 (6) |
| Secondary/high school | 160 (55.2) | 97 (26.9) | 257 (39.5) |
| College/university | 106 (36.6) | 248 (68.9) | 354 (54.5) |
| Employment status | | | |
| Not employed | 109 (37.7) | 81 (22.5) | 190 (29.3) |
| Employed | 180 (62.3) | 279 (77.5) | 459 (70.7) |
| Experienced COVID-19-related job loss | | | |
| No | 117 (40.3) | 167 (46.4) | 284 (43.7) |
| Yes | 173 (59.7) | 193 (53.6) | 366 (56.3) |
| Household water insecurity* (shortened HWISE scale) | | | |
| No | 206 (71.1) | 293 (81.4) | 499 (76.8) |
| Yes | 84 (28.9) | 67 (18.6) | 151 (23.2) |
| Household water insecurity dimensions experienced sometimes or often* | | | |
| Worry about meeting household water needs | 66 (22.8) | 38 (10.6) | 104 (16.0) |
| Going without washing hands after dirty activities because of problems with water | 48 (16.6) | 56 (15.6) | 104 (16.0) |
| No useable or drinkable water whatsoever in your household | 48 (16.6) | 48 (13.3) | 96 (14.8) |
| Problems with water caused you or anyone in your household to feel ashamed, excluded, stigmatized | 36 (12.4) | 25 (6.9) | 61 (9.4) |
| Self-reported HIV serostatus | | | |
| HIV negative | 265 (91.4) | 337 (93.6) | 602 (92.6) |
| HIV positive | 25 (8.6) | 23 (6.4) | 48 (7.4) |
| Anxiety symptoms (screened with Generalized anxiety disorder [GAD-2]) | | | |
| No | 203 (70) | 262 (72.8) | 465 (71.5) |
| Yes | 87 (30) | 98 (27.2) | 185 (28.5) |
| Depression symptoms (screened with Patient Health Questionnaire [PHQ-2]) | | | |
| No | 197 (67.9) | 267 (74.2) | 464 (71.4) |
| Yes | 93 (32.1) | 93 (25.8) | 186 (28.6) |

*(Continued)*

**Table 1.** (*Continued*)

| Characteristics | Mumbai: n = 290 | Bangkok: n = 360 | Total: N = 650 |
|---|---|---|---|
| | N (%) | N (%) | N (%) |
| Age (Mean/SD) | 29.8 (7.1) | 31.5 (7.8) | 30.7 (7.5) |
| Loneliness (assessed with Three-Item Loneliness Scale) | | | |
| No | 132 (45.5) | 173 (48.1) | 305 (46.9) |
| Yes | 158 (54.5) | 187 (51.9) | 345 (53.1) |
| Alcohol misuse (screened with AUDIT-C) | | | |
| No | 229 (79.0) | 217 (60.3) | 446 (68.6) |
| Yes | 61 (21.0) | 143 (39.7) | 204 (31.4) |
| COVID stress score (screened with COVID-19 Stress Scale) (mean, SD) | 17.2 (7.4) | 20.7 (7.2) | 19.1 (7.5) |
| Resilience score (assessed with the RSES-4 Scale) (mean, SD) | 13.7 (4.2) | 15.3 (3.2) | 14.6 (3.8) |

*in the past 4 weeks.

19-related job loss (56.3%). There were location differences across variables. Compared with the sample from Bangkok, those from Mumbai reported lower education levels (less likely to have completed college/university), and higher unemployment, household water insecurity and alcohol misuse. (Table 1) These multiple significant differences in the samples, alongside different sociocultural contexts, geographies, national LGBT rights and acceptance, national gross domestic product and underlying water insecurity prevalence, signal the importance of identifying country-specific data to inform tailored research, policy and practice. Thus, we conducted data analyses separately by site.

### Binary and multivariate logistic regression modeling of water insecurity among LGBTQ peoples in Mumbai and Bangkok

Unadjusted and adjusted multinomial logistic regression results are displayed in Tables 2–5.

Sociodemographic factors associated with water insecurity in Mumbai include identifying as transgender versus cisgender (aOR=3.35, 95% CI: 1.90–5.92) and experiencing COVID-19-related job loss (aOR=2.52, 95% CI: 1.41–4.53). In Bangkok, higher education was associated with lower odds of water insecurity (aOR=0.14, 95% CI: 0.05, 0.44). (Table 2)

Participants in both locations who reported water insecurity were more likely to experience depression (Mumbai: aOR=2.10, 95% CI: 1.19–3.69; Bangkok: aOR=2.14, 95% CI: 1.17–3.89) and anxiety (Mumbai: aOR=2.19, 95% CI: 1.23–3.88; Bangkok: aOR=1.99, 95% CI: 1.11–3.57) symptoms compared to those who did not report water insecurity (Table 3).

Alcohol misuse was twice as likely for participants in both locations (Mumbai: aOR=2.00, 95% CI: 1.04–3.83; Bangkok: aOR=2.06, 95% CI: 1.16–3.66) who experienced water insecurity compared to those who did not. Additionally, participants who experienced water insecurity reported significantly higher COVID-19 stress scores (Mumbai: β=3.86, 95% CI: 1.96, 5.76; Bangkok: β=4.03, 95% CI: 2.15, 5.92) than water secure counterparts (Table 4).

In Mumbai, water insecurity was associated with lower resilience (β =−1.36, 95% CI: −2.44, −0.28) when coping with stressful experiences, and participants in both locations who experienced water insecurity were more likely to report loneliness (aOR=2.50, 95%CI: 1.41–4.43; Bangkok: aOR=2.49, 95%CI: 1.36–4.57) than water secure counterparts. (Table 5)

### Discussion

Our study identifies multiple mental health outcomes – including nearly two-fold higher rates of anxiety and depression symptoms, loneliness, alcohol misuse and COVID-19-related stress – associated with water insecurity among LGBTQ participants in India and Thailand. This aligns with global literature on the complex direct and indirect pathways from water insecurity to mental health among cisgender and heterosexual populations (Wutich et al., 2020) and signals the urgent need to consider LGBTQ persons' experiences and health concerns related to water insecurity. As LGBTQ persons are disproportionately impacted by mental health concerns due to stigma and social marginalization (Ayhan et al., 2020; Mongelli et al., 2019; Scheim et al., 2020; Williams et al., 2021), we identify water insecurity as a relatively commonplace experience among LGBTQ persons in Mumbai and Thailand, which may operates as an additional stressor that warrants further investigation and intervention.

Our findings build on global literature with cisgender women in urban LMIC settings, including Odisha, India (Hulland et al., 2015) and Kathmandu Valley, Nepal, which showed linkages between water insecurity and depression (Aihara et al., 2016) in three key ways. First, we show that these associations exist for LGBTQ persons in other urban LMIC contexts (Bangkok, Thailand). Second, we demonstrate that water insecurity is associated with a range of mental health challenges, including but extending beyond depression – encompassing intrapersonal (anxiety, alcohol misuse), social (loneliness) and coping (resilience) dimensions of mental well-being. Third, the findings suggest differences in experiences within and between LGBTQ communities, whereby water insecurity most affected transgender persons in Mumbai compared to cisgender counterparts, those with lower education in Bangkok and COVID-19 job loss in Mumbai, and participants in India versus Thailand. This reflects the need articulated by Brewis et al. (2023) and Caruso et al. (2022) for an intersectional analysis on water-related insecurity that accounts for sexual and gender diversity as well as other socioeconomic and contextual factors.

LGBTQ persons in LMIC may have distinct water needs, as they may be at the nexus of LGBTQ stigma-related barriers to accessing WASH (Heller n.d.) *and* water-insecure living conditions. For instance, LGBTQ-related stigma may result in the denial of basic needs such as drinking water and hygiene kits in emergency situations (Wolf, 2019) and constrained access to sanitation services at large (Heller, n.d.; Brewis et al., 2023; Farber, 2023), while LGBTQ persons are also being more likely to live in slums and informal settlements with limited WASH access (Goel, 2016, 2020; Newman et al., 2021; Reid et al., 2022). Thus, LGBTQ-related social marginalization can constrain WASH access in work, community, healthcare and home environments, in turn producing water-related stress – this may interact with and exacerbate distal minority stress processes (e.g., violence, discrimination) and proximal minority stress processes (e.g., rejection expectation, identity concealment) central to the minority stress model's conceptualization of how minority stress harms LGBTQ persons' mental health (Meyer, 2003). While pathways from water insecurity can directly affect mental health via psychosocial stress and worry, future research with LGBTQ persons could also study indirect pathways identified

**Table 2.** Sociodemographic factors associated with water security among participants in Mumbai and Bangkok (*N* = 650)

| | Mumbai, India (*n* = 290) | | | | Bangkok, Thailand (*n* = 360) | | | |
|---|---|---|---|---|---|---|---|---|
| | Water insecurity | | | | Water insecurity | | | |
| | No | Yes | Odds ratio/95% CI | | No | Yes | Odds ratio/95% CI | |
| Characteristics | *N* (%) | *N* (%) | Crude | Adjusted | *N* (%) | *N* (%) | Crude | Adjusted |
| Age | 30.1 (7.2) | 29.0 (6.6) | 0.98 (0.94, 1.02) | 0.97 (0.93, 1.02) | 31.4 (7.9) | 31.9 (7.2) | 1.01 (0.97, 1.04) | 1.01 (0.98, 1.05) |
| Gender | | | | | | | | |
| Cisgender | 139 (79.9) | 35 (20.1) | Ref | Ref | 202 (83.1) | 41 (16.9) | Ref | Ref |
| Transgender | 67 (57.8) | 49 (42.2) | 2.90 (1.72, 4.90)** | 3.35 (1.90, 5.92)** | 91 (77.8) | 26 (22.2) | 1.41 (0.81, 2.44) | 1.21 (0.68, 2.16) |
| Education status | | | | | | | | |
| None/primary/elementary | 17 (70.8) | 7 (29.2) | Ref | Ref | 7 (46.7) | 8 (53.3) | Ref | Ref |
| Secondary/high school | 112 (70.0) | 48 (30.0) | 1.04 (0.41, 2.67) | 1.23 (0.45, 3.33) | 71 (73.2) | 26 (26.8) | 0.32 (0.11, 0.97)* | 0.29 (0.09, 0.92)* |
| College/university | 77 (72.6) | 29 (27.4) | 0.91 (0.34, 2.43) | 1.53 (0.53, 4.42) | 215 (86.7) | 33 (13.3) | 0.13 (0.05, 0.39)** | 0.14 (0.05, 0.44)** |
| Employment status | | | | | | | | |
| No | 72 (66.1) | 37 (33.9) | Ref | Ref | 61 (75.3) | 20 (24.7) | Ref | Ref |
| Yes | 133 (73.9) | 47 (26.1) | 0.69 (0.41, 1.15) | 0.88 (0.50, 1.54) | 232 (83.2) | 47(16.9) | 0.62 (0.34, 1.12) | 0.87 (0.46, 1.66) |
| HIV status | | | | | | | | |
| Negative | 187 (70.6) | 78(29.4) | Ref | Ref | 273 (81.1) | 64 (18.9) | Ref | Ref |
| Positive | 19 (76.0) | 6 (24.0) | 0.76 (0.29, 1.97) | 0.90 (0.33, 2.47) | 20 (86.9) | 3 (13.1) | 0.64 (0.18, 2.22) | 0.59 (0.16, 2.11) |
| COVID-19 job loss | | | | | | | | |
| No | 159 (69.7) | 69 (30.3) | Ref | Ref | 47 (67.1) | 23 (32.9) | Ref | Ref |
| Yes | 47 (75.8) | 15 (24.2) | 2.23 (1.28, 3.87)** | 2.52 (1.41, 4.53)** | 246 (84.8) | 44 (15.2) | 2 (1.15, 3.50)* | 1.71 (0.94, 3.12) |

CI, confidence interval.
*$p < 0.05$;
**$p < 0.01$.

**Table 3.** Water insecurity and sociodemographic factors associated with depression symptoms and anxiety symptoms among study participants in Mumbai, India and Bangkok, Thailand (*N* = 650)

| | Depression symptoms | | | | Anxiety symptoms | | | |
| --- | --- | --- | --- | --- | --- | --- | --- | --- |
| | Mumbai, India | | Bangkok, Thailand | | Mumbai, India | | Bangkok, Thailand | |
| Characteristics | Crude OR (95% CI) | Adjusted OR (95% CI) | Crude OR (95% CI) | Adjusted OR (95% CI) | Crude OR (95% CI) | Adjusted OR (95% CI) | Crude OR (95% CI) | Adjusted OR (95% CI) |
| Age | 0.97 (0.94, 1.01) | 0.98 (0.94, 1.01) | 1.00 (0.97, 1.03) | 1.00 (0.97, 1.03) | 1.02 (0.99, 1.06) | 1.03 (0.99, 1.06) | 1.01 (0.98, 1.04) | 1.01 (0.98, 1.04) |
| Gender | | | | | | | | |
| Cisgender | Ref | Ref | Ref | Ref | Ref | Ref | Ref | Ref |
| Transgender | 1.13 (0.68, 1.86) | 0.84(0.48, 1.47) | 1.20 (0.73, 1.97) | 1.15 (0.68, 1.94) | 1.42 (0.86, 2.36) | 1.10 (0.63, 1.93) | 0.89 (0.54, 1.46) | 0.84 (0.50, 1.41) |
| Water insecure | | | | | | | | |
| No | Ref | Ref | Ref | Ref | Ref | Ref | Ref | Ref |
| Yes | 2.11(1.24, 3.58)** | 2.10 (1.19, 3.69)* | 2.32 (1.33, 4.06)** | 2.14 (1.17,3.89)* | 2.12 (1.24, 3.61)** | 2.19 (1.23, 3.88)** | 2.11 (1.21, 3.68)** | 1.99 (1.11, 3.57)* |
| Educational status | | | | | | | | |
| None/primary/ elementary | Ref | Ref | Ref | Ref | Ref | Ref | Ref | Ref |
| Secondary/high school | 0.59 (0.24, 1.39) | 0.55 (0.22, 1.34) | 0.29 (0.09, 0.92)* | 0.29 (0.09, 0.96)* | 0.62 (0.26, 1.49) | 0.61 (0.25, 1.51) | 0.38 (0.12, 1.14) | 0.43 (0.14, 1.34) |
| College/university | 0.45 (0.18, 1.11) | 0.42 (0.16, 1.09) | 0.41 (0.14, 1.19) | 0.61 (0.20, 1.85) | 0.50 (0.20, 1.26) | 0.50 (0.19, 1.32) | 0.42 (0.15, 1.21) | 0.57 (0.19, 1.69) |
| Job lost | | | | | | | | |
| No | Ref | Ref | Ref | Ref | Ref | Ref | Ref | Ref |
| Yes | 1.35 (0.81, 2.25) | 1.20 (0.70, 2.04) | 2.69 (1.62. 4.48)* | 2.84 (1.68, 4.83)* | 1.01 (0.60, 1.68) | 0.89 (0.52, 1.52) | 1.45 (0.90, 2.32) | 1.43 (0.88, 2.34) |

CI, confidence interval.
*$p < 0.05$;
**$p < 0.01$.

**Table 4.** Water insecurity and sociodemographic factors associated with alcohol misuse and COVID-19 stress among participants in Mumbai, India and Bangkok, Thailand (*N* = 650)

| | Alcohol misuse symptoms | | | | COVID-19 stress | | | |
|---|---|---|---|---|---|---|---|---|
| | Mumbai, India | | Bangkok, Thailand | | Mumbai, India | | Bangkok, Thailand | |
| Characteristics | Crude OR (95% CI) | Adjusted OR (95% CI) | Crude OR (95% CI) | Adjusted OR (95% CI) | Crude OR (95% CI) | Adjusted OR (95% CI) | Crude OR (95% CI) | Adjusted OR (95% CI) |
| Age | 1.04 (0.99, 1.08) | 1.04(1.00, 1.08)* | 0.96(0.93, 0.98)* | 0.96(0.93, 0.99)* | 0.1 (−0.02, 0.22) | 0.12 (−0.01, 0.24)* | −0.02 (−0.11, 0.08) | −0.03(−0.12, 0.06) |
| Gender | | | | | | | | |
| Cisgender | Ref | Ref | Ref | Ref | Ref | Ref | Ref | Ref |
| Transgender | 1.46 (0.81, 2.61) | 1.28 (0.68, 2.42) | 1.21(0.76,1.92) | 1.13 (0.69, 1.84) | 0.80 (−0.95, 2.55) | 0.23 (−1.57, 2.03) | 0.94 (−0.64, 2.52) | 0.62 (−0.91, 2.15) |
| Water insecure | | | | | | | | |
| No | Ref | Ref | Ref | Ref | Ref | Ref | Ref | Ref |
| Yes | 1.91(1.04, 3.49)* | 2.00(1.04, 3.83)* | 2.34(1.36, 4.01)** | 2.06(1.16, 3.66)* | 4.19 (2.36, 6.01)** | 3.86 (1.96, 5.76)** | 4.46(2.61, 6.31)** | 4.03 (2.15, 5.92)** |
| Educational status | | | | | | | | |
| None/primary/elementary | Ref | Ref | Ref | Ref | Ref | Ref | Ref | Ref |
| Secondary/high school | 1.30 (0.42, 4.06) | 1.34 (0.42, 4.29) | 1.03 (0.35, 3.07) | 1.25 (0.40, 3.90) | −0.19 (−3.39, 3.0) | −0.27 (−3.35, 2.80) | −2.94 (−6.83, 0.96) | −2.24 (−6.00, 1.53) |
| College/university | 1.16 (0.36, 3.78) | 1.24 (0.36, 4.22) | 0.41 (0.14, 1.19) | 0.55 (0.18, 1.68) | 0.12(−3.18, 3.43) | 0.17 (−3.08, 3.41) | −3.01(−6.75, 0.72) | −0.96 (−4.61, 2.69) |
| Job lost | | | | | | | | |
| No | Ref | Ref | Ref | Ref | Ref | Ref | Ref | Ref |
| Yes | 0.84 (0.47, 1.50) | 0.76 (0.41, 1.40) | 1.30 (0.84, 2.03) | 1.01 (0.63, 1.62) | 2.85 (1.13, 4.56)** | 2.26 (0.55, 3.97)** | 3.16(1.71, 4.62)** | 2.94 (1.47, 4.40)** |

CI, confidence interval.
*$p < 0.05$;
**$p < 0.01$.

**Table 5.** Water insecurity and sociodemographic factors associated with resilience and loneliness among participants in Mumbai, India and Bangkok, Thailand (*N* = 650)

| | Resilience | | | | Loneliness | | | |
|---|---|---|---|---|---|---|---|---|
| | Mumbai, India | | Bangkok, Thailand | | Mumbai, India | | Bangkok, Thailand | |
| Characteristics | Crude OR (95% CI) | Adjusted OR (95% CI) | Crude OR (95% CI) | Adjusted OR (95% CI) | Crude OR (95% CI) | Adjusted OR (95% CI) | Crude OR (95% CI) | Adjusted OR (95% CI) |
| Age | 0.04 (−0.03, 0.1) | 0.02(−0.04, 0.09) | 0.02 (−0.02, 0.06) | 0.02(−0.03, 0.06) | 0.99 (0.96−1.03) | 0.99 (0.96−1.03) | 0.99 (0.97−1.02) | 0.99 (0.96, 1.02) |
| Gender | | | | | | | | |
| Cisgender | Ref | Ref | Ref | Ref | Ref | Ref | Ref | Ref |
| Transgender | −0.93(−1.90, −0.05) | −0.09(−1.12, 0.93) | 0.12 (−0.59, 0.84) | 0.23(−0.49, 0.94) | 0.88 (0.55, 1.41) | 0.81 (0.48, 1.36) | 1.52 (0.97, 2.38) | 1.56 (0.98, 2.49) |
| Water insecure | | | | | | | | |
| No | Ref | Ref | Ref | Ref | Ref | Ref | Ref | Ref |
| Yes | −1.27(−2.32, −0.22)* | −1.36(−2.44, −0.28)* | −0.92(−1.77, −0.06)* | −0.63(−1.52, 0.25) | 2.36(1.38, 4.05)** | 2.50(1.41, 4.43)** | 2.57(1.45, 4.55)** | 2.49(1.36, 4.57)** |
| Educational status | | | | | | | | |
| None/primary/elementary | Ref | Ref | Ref | Ref | Ref | Ref | Ref | Ref |
| Secondary/high school | 2.04(0.29, 3.79)* | 2.00(0.25, 3.75)* | 1.39 (−0.34, 3.14) | 1.23(−0.53, 2.99) | 1.00 (0.42, 2.36) | 0.95 (0.39, 2.29) | 0.19 (0.05, 0.72)* | 0.20(0.05, 0.79)* |
| College/university | 3.26(1.45, 5.07)** | 3.18 (1.34, 5.02)** | 2.18 (0.51, 3.85)* | 1.90(0.19, 3.61)* | 1.65 (0.68, 4.02) | 1.57 (0.62, 4.01) | 0.29 (0.08, 1.05) | 0.42 (0.11, 1.61) |
| Job lost | | | | | | | | |
| No | Ref | Ref | Ref | Ref | Ref | Ref | Ref | Ref |
| Yes | 0.73(−0.24, 1.71) | 0.97(−0.002, 1.94) | −0.37 (−1.04, 0.30) | −0.18(−0.87, 0.51) | 1.32 (0.82, 2.11) | 1.16 (0.71, 1.90) | 1.62 (1.07, 2.46)* | 1.69(1.08, 2.64)* |

CI, confidence interval.
*$p < 0.05$;
**$p < 0.01$.

in other populations and contexts, such as food insecurity and sanitation insecurity (Brewis et al., 2019) and stigma linked with insufficient water access (Young et al., 2019). Due to structural violence in education and employment (Chakrapani et al., 2007), LGBTQ persons are overrepresented in sex work (Platt et al., 2022), including in India (Chakrapani et al., 2022a, 2022b) and Thailand (Farber, 2023; Newman et al., 2021; Ojanen et al., 2019, 2019; Reid et al., 2022). Sex work is itself an occupation that requires WASH access for optimal health (Grittner and Sitter, 2019; Sherman et al., 2023); thus, sex workers (regardless of LGBTQ identity) have occupational WASH needs that require additional attention.

This study is among the first to examine water insecurity and linkages with mental health among LGBTQ persons. However, there are several study limitations. Due to the non random sample, findings are not generalizable to all LGBTQ persons in Mumbai and Bangkok, and the cross-sectional design precludes determining causality. It is not possible with the study design to assess if there is differential risk for water insecurity among LGBTQ persons versus non-LGBTQ persons in each site. The small sample size did not allow us to examine sexual orientation differences to better characterize linkages between water insecurity and mental health. The online recruitment strategy itself has limitations because, as discussed throughout this manuscript, LGBTQ persons may be more likely to live in poverty than heterosexual and cisgender counterparts, and in turn may experience more barriers to accessing mobile devices and internet data. Hence, the online recruitment strategy could potentially miss LGBTQ persons who are the most marginalized and most likely to face water insecurity, reflecting the digital divide of inequitable access to technology and in turn inequitable access to digital clinical trials (Wirtz et al., 2022). In fact, if the most marginalized are not included in this study, the issues of water insecurity among LGBTQ persons may be larger than what we have measured. Despite these limitations, this study identifies water insecurity as a future action point for research, policy and practice with LGBTQ populations in Bangkok and Mumbai.

## Conclusion

To date, water insecurity has been overlooked in LGBTQ mental health research, and LGBTQ persons have largely been overlooked in water insecurity research. Our study is unique in identifying the prevalence and sociodemographic factors associated with water insecurity with a sample of LGBTQ persons in two urban LMIC settings, as well as spotlighting a wide range of mental health challenges associated with water insecurity. With ongoing LGBTQ urbanization, increases in climate-related water insecurity, preexisting LGBTQ social marginalization and health inequities, researchers can start including sexual orientation and gender identity in WASH research and water insecurity in mental health research with LGBTQ persons. Water insecurity has been described as syndemic with mental health and other social inequities (Workman and Ureksoy, 2017). Thus, mental health interventions with LGBTQ persons in water-insecure contexts could address co-occurring social (e.g., isolation, stigma), structural (e.g., poverty, water insecurity) and health (e.g., depression) inequities to advance health and rights.

**Open peer review.** To view the open peer review materials for this article, please visit http://doi.org/10.1017/gmh.2024.27.

**Data availability statement.** Data available upon reasonable request from Peter A. Newman (p.newman@utoronto.ca) and upon obtaining required research ethics approvals.

**Acknowledgments.** The authors would like to acknowledge lead community partners, the Humsafar Trust (Mumbai), the Institute for HIV Research and Innovation (Bangkok) and VOICES-Thailand (Bangkok), for their input and collaboration in implementing the study. The authors would also like to thank community-based organization and clinic staff, and their dedicated peer counselors, many of whom are LGBT+ individuals. The authors would like to thank the reviewers and Editor for their valuable contributions and feedback.

**Author contribution.** C.H.L.: Conceptualization of paper and analysis, methodology, led writing (original draft preparation). P.A.N.: Conceptualization of study, methodology, investigation, data curation, writing-reviewing and editing. Z.A.: Data analysis, contributed to writing and editing. F.M.: Contributed to writing and editing. V.C.: Investigation, data curation, contributed to writing, reviewing and editing. S.T.: Investigation, data curation, contributed to writing and editing. M.S.: Investigation, data curation, contributed to writing and editing. P.A.: Investigation, data curation, contributed to writing and editing.

**Financial support.** This study is funded by IDRC (International Development Research Centre, Canada; #109555) (PI: Newman). Logie's, Admassu's and MacKenzie's efforts were in part supported by Logie's funding from the Canada Research Chairs Program (CRC Tier 2 in Global Health Equity and Social Justice with Marginalized Populations).

**Competing interest.** The authors declare none.

**Ethics statement.** Ethical approval was obtained from the Research Ethics Boards of the University of Toronto (39769), the Humsafar Trust (HST-IRB-S1-06/2021) and Chulalongkorn University (272/64). The trial is registered at ClinicalTrials.gov NCT04870723.

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
