## [Editor Report]

Thank you for this manuscript, which begins to elucidate and connect several interrelated concepts: mental health, sexual and gender minority populations, and water insecurity.

I concur with Reviewer 1, especially in requesting that a bit more attention be paid to the relationship between LGBTQ+ identity and water insecurity. Understanding, as well, that this was a cross-sectional study (with no comparison group), many readers will anticipate a deeper exploration/explanation of how LGBTQ+ groups may be different than other people in the same settings where water is scarce. The discussion may, therefore, benefit from more explicit reasoning. As Reviewer 1 states, “My overarching question/concern is this: I recognize that the study design cannot determine causation, but I am not sure if the authors are claiming that LGBTQ people have specific water needs that are not being met, or that their position in marginalized communities is associated with water insecurity (but that their insecurity might be an effect of being in communities that are economically precarious, inter alia, as opposed to a direct association).”

If you feel like you can address that overarching issue, I invite a manuscript revision.

Many thanks,

---

## [Editor Report]

Dear Authors - this is a great revision and I am prepared to recommend this manuscript for publication; however, Reviewer 2 had a few remaining minor points that I invite you to address first.